# Heavy Metals in Muscle Tissue of *Pterois volitans* from the Veracruz Reef System National Park, Mexico

**DOI:** 10.3390/ijerph16234611

**Published:** 2019-11-20

**Authors:** Jesús Montoya-Mendoza, Estrella Alarcón-Reyes, María del Refugio Castañeda-Chávez, Fabiola Lango-Reynoso, Rosa E. Zamudio-Alemán

**Affiliations:** Laboratorio de Investigación en Recursos Acuáticos, Tecnológico Nacional de México, Instituto Tecnológico de Boca del Río, Carretera Veracruz-Córdoba km 12, Boca del Río, 94290 Veracruz, Mexico; estrella@bdelrio.tecnm.mx (E.A.-R.); mariacastaneda@bdelrio.tecnm.mx (M.d.R.C.-C.); fabiolalango@bdelrio.tecnm.mx (F.L.-R.); rosazamudio@bdelrio.tecnm.mx (R.E.Z.-A.)

**Keywords:** reef, trace metals, lionfish

## Abstract

Concentrations of cadmium (Cd), lead (Pb), vanadium (V), and zinc (Zn) were measured in the muscle of 30 specimens of *Pterois volitans*, captured on April 2018, in the Veracruz Reef System National Park (VRSNP), Veracruz, Mexico. Concentrations, in the samples, were quantified with atomic absorption spectrophotometry (AAS), after microwave digestion. Results of the mean concentration, in descending order were V = 7.3 ± 0.7; Pb = 0.66 ± 0.07; Zn = 0.43 ± 0.14; and Cd = 0.03 ± 0.01 mg kg^−1^ dry weight. These values did not exceeded limits established in the Mexican National Standard (NOM-242-SSA1-2009), of Cd and Pb (0.5 mg kg^−1^) wet weight. This means that consumption of lionfish from this site does not pose a potential risk for human health.

## 1. Introduction

The Veracruz Reef System National Park (VRSNP) is located in an area with high urban, agricultural, and industrial development, signifying the presence of multiple contaminant sources for the marine environment. It also has a strong continental influence from discharges of the rivers La Antigua, Jamapa, and Papaloapan, carrying a number of chemical substances [1]. Wastewater discharges have been detected in the Jamapa River [2,3]; agricultural discharges in La Antigua river [4]; and in the Papaloapan these discharges come from different anthropogenic activities [5]. Among the top contaminants damaging this coastal environment are heavy metals (HMs) [6]. The presence of HMs has been detected in marine sediments of the VRSNP, for example Fe, V, Cr, Zn, Ni, Pb, and Cu, [7]; and Cu, Cd, and Zn [2]. HMs such as Cd, Cu, and Pb have been found in benthic macroalgae [8]. On the other hand, lionfish, *Pterois volitans*, whose natural distribution is in the Indo-Pacific region [9], was registered in the Anegada de Adentro reef of the VRSNP, in 2012 [10]. This organism represents an ecological problem because it is an invasive species that affects local fish populations and is considered a top consumer in places it has invaded, due to the lack of a natural predator. Lionfish could behave as an indicator species and would allow the assessment of pollution levels in the marine environment [11]. Various heavy metals have been detected in muscle from specimens captured in Florida, USA [12,13], Curacao [14], and Jamaica [15], these concentrations have not exceeded the maximum limits allowed by the World Health Organization (WHO) [16]. Since the VRSNP dministration is promoting the capture and consumption of lionfish as a strategy to reduce their populations, the objective of this study has been to analyze the concentration of Cd, Pb, V, and Zn in the muscle tissue of *P. volitans*, to determine whether it represents any risk to human health. 

## 2. Materials and Methods 

In April 2018, 30 specimens of *P. volitans* were captured in five reefs of the VRSNP, as follows: Pájaros Reef, n = 11 (19°11′18″N, 96°05′22″W); Isla Sacrificios, n = 2 (19°10′35″N, 96°05′30″W); Isla Verde, n = 5 (19°12′09″N, 96°03′58′’W); Enmedio Reef, n = 6 (19°06′21″N, 95°56′18″W); and Anegada de Afuera island, n = 6 (19°09′25″N, 95°51′15″W) (Figure 1). Capture was made with lance harpoons and basic equipment by local fishers. Fish were transported in airtight bags into an ice cooler at ±4 °C to the Laboratorio de Investigación en Recursos Acuáticos in the facilities of the Instituto Tecnológico de Boca del Río. The study was conducted under the Declaration of Helsinki, and the Ethics Committee of Programa para el Desarrollo Profesional Docente, para el Tipo Superior (PRODEP) approved the protocol (project 511-6/18-10343).

Sample processing followed the standards to minimize cross-contamination risks [16,17]. Total length was registered in centimeters using an ictiometer and total weight in grams with an analytical scale (AE-ADAM, Oxford, UK). A sample of skinless muscle was taken from the lateral part of the fish. Cuts were made with scalpel and steel blades and placed into airtight bags. Samples were processed with the drying, grinding, and digestion technique recommended by the NOM-117-SSA1-1994 [18]. For the drying part, 10 g of sample was taken in porcelain capsules that were placed in an oven, at 60 °C for a week, until reaching constant weight. Dry samples were grinded in a porcelain mortar to homogenization. For the digestion step, 0.5 g of each sample was placed in a Teflon beaker HP‑500, adding 10 mL of HNO_3_, reagent grade 70% (J.B. Baker, Xalostoc, México), and placed in a microwave oven, CEM‑Mars 5 model (CEM-Corporation, Matthews, NC, USA), adding one blank and one control sample. This process was conducted at 150 °C for 5 min and at 190 °C for 10 min, following the manufacturer recommendations (CEM Corporation). Then, samples were filtered in a Nalgene bottle; using 0.45 μm Millipore nitrocellulose membrane filters (MF-Millipore, Tullagreen, Ireland) in a vacuum pump V-700 (Buchi, Flawil, Switzerland). The filtered solution was diluted to the 50 mL mark with distilled water, and stored in refrigeration at ±4°C, in polyethylene bottles until reading. 

Readings of Cd, Pb, V, and Zn were made in an atomic absorption spectrophotometer, Thermo Scientific 3500 Model AA Ice System (Thermo Scientific, Jingzhou, China), fitted with a hollow cathode lamp, specific for the determination of each metal (detection limits: Cd = 0.0028, Pb = 0.016, V = 0.11, and Zn = 0.0033 mg kg^−1^). Certified standards, with correlation coefficient >0.99, were used to prepare the standard curve for each item tested.

The concentration of each metal was compared with limits established by NOM-242-SSA1-2009 [19] and with international recommendations of the Joint Food and Agriculture Organization (FAO) and WHO [16]. The normal distribution was confirmed with a Kolmogorov–Smirnov Test. The Pearson correlation index was analyzed between the Cd, Pb, V, and Zn concentrations in muscle tissue of *P. volitans* and the organism size, weight, and Fulton condition factor [20]. The variation of concentrations in all four metals in the sampling zone was also determined with an ANOVA (*p* < 0.05) [21]. Similarity of metal concentrations related to collection sites was calculated using Euclidean distances [22]. Statistical analyses were performed using the PAST software, version 3.14 [23].

## 3. Results and Discussion

Total length of the 30 *P. volitans* specimens ranged from 16 to 36 cm (mean 23 ± 5.6 cm), and weight ranged from 32 to 717 g (mean 236.2 ± 210 g). The total average ± SD metal concentration and minimum and maximum values were as follows: V = 7.3 ± 0.7 mg kg^−1^ (5.9–8.3); Pb = 0.66 ± 0.07 mg kg^−1^ (0.47–0.74); Zn = 0.43 ± 0.14 mg kg^–^¹ (0.3–0.8); and Cd = 0.03 ± 0.01 mg kg^−1^ (0.01–0.04). Concentration records for all four metals were adjusted to the normal distribution type (*p* > 0.05). In 100% of the samples, the Cd, V, Pb, and Zn concentrations were within allowable limits. Mean concentration records per collection site are displayed in Table 1. Finally, the Fulton condition factor for the sample was 1.93, showing that the fish had no stressing conditions. 

On the other hand, Mexican legislation in effect has no limits established for V and Zn [19]. Vanadium concentrations detected are lower than those established as effects range low (ERL); and Zn concentrations are below the apparent effects threshold (AET) in *Neanthes* sp., proposed by the National Oceanic and Atmospheric Administration (NOAA) [24]. Therefore, none of these metals poses a health risk. Now, Cd and Pb are identified as contaminant with limits of 0.5 mg kg^−1^ wet weight, according to Mexican standards and the international standard FAO/WHO established in the Codex Alimentarius 193-1995 (0.2 mg kg^−1^ wet weight in fish muscle) [16]. In this study, the average concentration of cadmium (0.007 mg kg^−1^ wet weight) and lead (0.15 mg kg^−1^ wet weight) did not exceed the established national and international limits. However, the presence of these contaminants and in particular of Pb in the lionfish muscle as a source of protein represents a risk to human health. 

It has to be noted that the regression analysis between the four metals concentrations with respect to length (Cd, r = 0.105; Pb, r = 0.213; V, r = 0.034; Zn, r = 0.172) and weight (Cd, r = 0.114; Pb, r = 0.250; V, r = 0.086; Zn, r = 0.152), and the Fulton condition factor (Cd, r = 0.33; Pb, r = 0.21; V, r = 0.29; Zn, r = 0.19), did not show a significant correlation even though contaminants present in the VRSNP are constant and interchangeable, because no pattern of contaminant distribution was observed. This was confirmed by the ANOVA analysis (*F* = 2.64, p = 0.058), which revealed no significant differences between the four metals and the collection sites. This situation was verified by the Student’s *t* test (α = 0.05), since it allows us to compare HM concentration means corresponding to the North or South location in the collection area (Table 1) and by the clusters formed by the similarity index using Euclidean distances, because these do not reveal any pattern of greater or lesser amount of metals by the North–South sampling sites or vice versa (Figure 2). 

Through this behavior, it can be observed that contaminants are spread across the reef park. Their presence can be explained by the permanent outflows of La Antigua river in the North, the Papaloapan river in the South, and the Jamapa river in the central part; the storm water discharges on the coast line near the reefs [25,26]; and urban wastewater streams from local treatment plants [27]. All these streams are subject to marine currents near the coastline [28], these currents apparently drag pollutants continuously between the reefs of the VRSNP [29]. Now, given the serious impacts caused by the invasive nature of lionfish, it is important to continue studying their distribution and abundance [30], so that in the field of environmental management [31], it becomes possible to develop control strategies based on the fish capture and consumption. 

Lionfish consumption has been promoted as protein source because, theoretically, it poses no risks caused by the HM presence [15], and the present study corroborates that the human consumption of this fish from the PNSAV does not represent such a risk. However, this study recorded Pb concentrations close to the international limits (0.11–0.17 mg kg^−1^ wet weight) [16], which are the cause of serious public health problems [16]. Therefore, it is necessary to consider the consumption frequency of this organism, to remain within the provisional tolerable weekly intake limits (PTWI) for Pb (25 µg kg^−1^ of body weight), recommended by the Joint FAO/WHO [32,33]. Taking into account those in Mexico, where the country mean body weight is 75 kg and the annual per capita fish fillet consumption is 230 g [34] the Pb estimated intake would be 2.38 µg kg^−1^, well below the Codex PTWI criteria of 25 µg Pb/kg^−1^ of body weight [33]. Similar results were given for the consumption of *Carcharhinus limbatus* from the northern Coast of Veracruz, Mexico [35], *Ariopsis felis* from the southern Gulf of Mexico, as well as *Scomberomorus cavalla* from the northern Gulf of Mexico, USA [36], and *Octopus vulgaris* from Southern Italy [37]. It should be noted, however, that the consumption of lionfish protein is a recent practice and not frequent, as it is an invasive species. More extensive work will be necessary to determine the risk posed by other contaminants. Soon the analysis of Hg and Cr in *P. volitans* of the PNSAV and other heavy metals in specimens of the Mexican Caribbean will be done. 

## 4. Conclusions

The presence of HMs was detected in muscle tissue of *P. volitans* from the VRSNP. Mean Cd and Pb concentrations were below the allowable limit. The levels of V and Zn in lionfish muscle were lower than those reported in the area. No correlation was observed among mean Cd, Pb, V, or Zn concentrations in muscle and the organisms’ size, weight, or condition factor. The availability of heavy metals for organisms inhabiting the VRSNP reefs may be due to the proximity to the coastline and the marine circulation patterns. Consumption of *P. volitans* protein from this area does not imply a potential risk to human health. 

## Figures and Tables

**Figure 1 ijerph-16-04611-f001:**
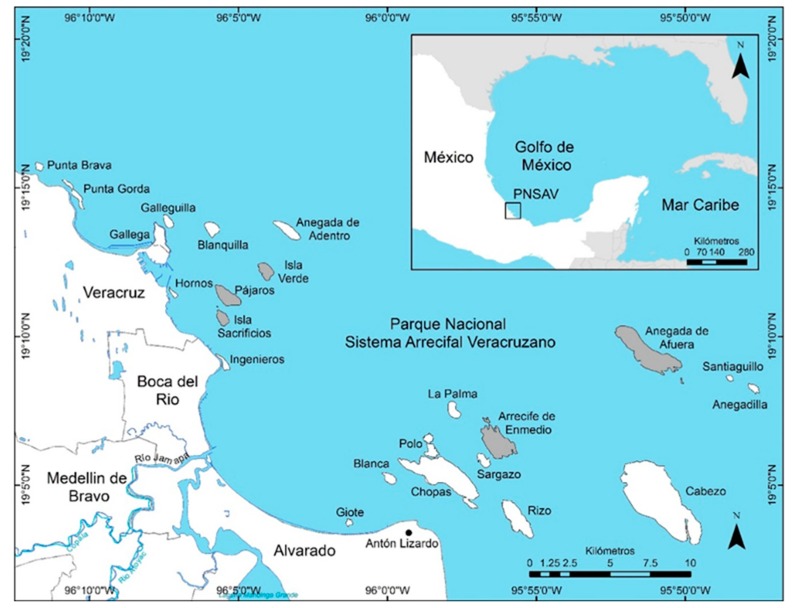
Geographical location of the Veracruz Reef System National Park. Sampling areas are shaded.

**Figure 2 ijerph-16-04611-f002:**
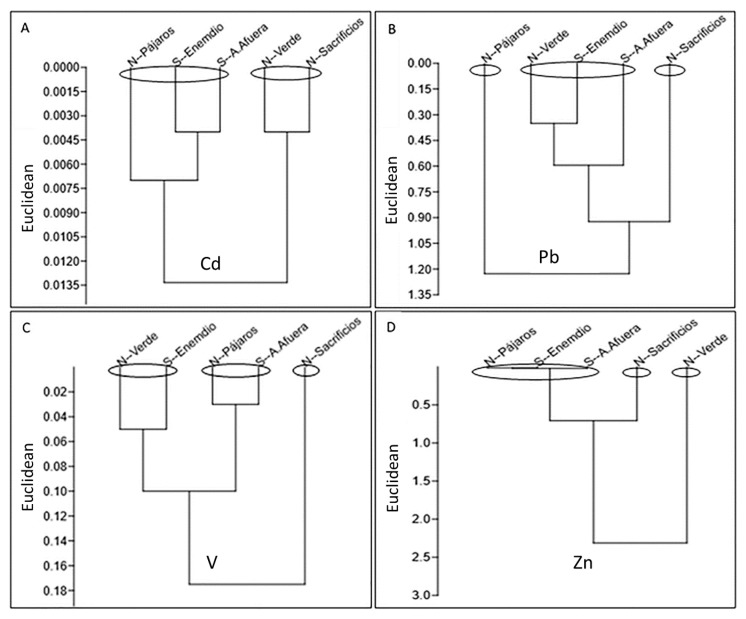
Similarity of heavy metal concentrations in *P. volitans* muscle tissue, in North (N) and South (S) reefs of the Veracruz Reef System National Park. Cluster of similarity (**A**, cadmium), (**B**, lead), (**C**, vanadium), (**D**, zinc).

**Table 1 ijerph-16-04611-t001:** Heavy metals detected in muscle of *Pterois volitans* from Veracruz Reef System National Park (VRSNP), Mexico.

Area		Mean Concentration, Dry Weight (±SD) (mg kg^−1^)			
Reef	n	Cd (0.5 mg kg^−1^) * (0.05 mg kg^−1^) **	Pb (0.5 mg kg^−1^) * (0.2 mg kg^−1^) **	V	Zn
North					
Pájaros	11	0.033 ± 0.010	0.72 ± 0.02	8.0 ± 0.2	0.37 ± 0.09
Isla de Sacrificios	2	0.012 ± 0.007	0.49 ± 0.03	6.1 ± 0.2	0.60 ± 0.4
Isla Verde	5	0.016 ± 0.008	0.59 ± 0.02	6.7 ± 0.3	0.67 ± 0.14
South					
Enmedio	6	0.024 ± 0.009	0.63 ± 0.03	7.0 ± 0.1	0.38 ± 0.02
Anegada de Afuera	6	0.027 ± 0.010	0.69 ± 0.01	7.4 ± 0.1	0.36 ± 0.02

SD, standard deviation. * Maximum limit wet weight (Mexican legislation) [19]. ** Maximum limit wet weight (WHO) [16].

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
