# Peer review of "Heavy Metals in Muscle Tissue of Pterois volitans from the Veracruz Reef System National Park, Mexico"

_ijerph, 2019, doi:10.3390/ijerph16234611_

Round 1

Reviewer 1 Report

I think it's a very interesting and very important topic for public health nowadays. I believe that this work needs low changes and improvements in introduction and in the conclusion section.

The lionfish is considered a resident fish species, and your findings could be compared to results of Ariano et al. 2019 (Metal concentration in muscle and digestive gland of common octopus (Octopus vulgaris) from two coastal site in Southern Tyrrhenian Sea (Italy) - Molecules 24, Issue 13, 2019) in the O. vulgaris.

Methods are well explained 

LINE 55 -  please use subscript for 3 of  HNO3

Author Response

Response to Reviewer 1 Comments

Point 1: I think it's a very interesting and very important topic for public health nowadays. I believe that this work needs low changes and improvements in introduction and in the conclusion section.

Response 1. Dear Reviewer, thank you for your comments and observations. The suggested changes were made in the introduction and conclusions sections.

Point 2: The lionfish is considered a resident fish species, and your findings could be compared to results of Ariano et al. 2019 (Metal concentration in muscle and digestive gland of common octopus (Octopus vulgaris) from two coastal site in Southern Tyrrhenian Sea (Italy) - Molecules 24, Issue 13, 2019) in the O. vulgaris.

Response 2:

We appreciate the recommendation and incorporate in the discussion, the information of this work and others

Point 3: Methods are well explained.

Response 3.

Thanks.

Point 4: LINE 55 -  please use subscript for 3 of  HNO3.

Response 4.

As suggested, HNO3.

.

Reviewer 2 Report

comments in text

Author Response

Response to Reviewer 2 Comments

Point 1: insert space

Response 1. Thanks for the comments and observations, and we insert space in, line 14, 16, 34, 80, 81, 88, 90, 120

Point 2: Line 44, Ethical permit

Response 2: The protocol was approved by the Ethics Committee of PRODEP (Project 511-6/18-10343).

Point 3: Provide manufacturer name, city, state, country.

Response 3. We add data requested of the line 55, 56, 57, 59, 60, 64

Point 4: LINE 64-68 - I suggest shortening the text.

Response 4.  We rewrote the paragraph as suggested.

.

Point 5: LINE 68 - Please provide detection limits, sensitivity, and calibration curves

Response 5.  We add the requested data.

Point 6: LINE 71 - insert references

Response 6.  We add references in the lines 71, 90, 119

Point 7: To correct, line 55, 57, 59, 91, 102, 103, 131, 146, 148, 180, 183, 184, 186, 194, 198, 211, 215.

Response 7.  The corrections indicated were made

Reviewer 3 Report

The MS "Heavy metals in muscle tissue of Pterois volitans, PNSAV, Mexico" describe the presence of Cd, Pb, V and Zn of Pterois volitans samples in marine environments of Mexico. The novelty of the paper relies on the first epidemiological data on the presence of these heavy metals in this organism revealing itself a pioneering study.

However, the present work presents several weaknesses regarding the data analysis. Several improvements regarding the method of investigation used and the discussion of the results are needed. All my suggestions are listed below:

Title

The acronyms PNSAV must be deleted, give the exact citation of the sampling area.

Abstract

Line 13: AA must give the SD of the average value.

Introduction

Line 35: Specify which metals were examined in this work.

Materials and Methods

Page 2 line 48: AA must specify the material of the scalpel used.

Page 2 line 51: Give the country of production of the balance

Page 3 line 74: Did the data obtained respected the assumption of normality of distribution and homoschedasticity?

Results and Discussion

Page 3 line 82: The measurement unit should be reported even for the concentration range.

A percentage of the fish samples above the limits should be given.

Page 3 lines 93-94: Regarding the correlation I suggest to use the Fulton condition factor.

Page 4 line 114: A comparison with other benthic fishes could be useful to have an epidemiological comparison. I suggest https://doi.org/10.1007/s00128-007-9028-7

Author Response

Response to Reviewer 3 Comments

Point 1: However, the present work presents several weaknesses regarding the dataanalysis. Several improvements regarding the method of investigation used and the discussion of the results are needed. All my suggestions are listed below:

Response 1. Thanks for the comments and observations, and we rewrite the methods and discussion sections

Point 2: Title, The acronyms PNSAV must be deleted, give the exact citation of the sampling area.

Response 2: Add sampling site

Point 3: Line 13: AA must give the SD of the average value.

Response 3. Add SD.

Point 4: Line 35: Specify which metals were examined in this work.

Response 4. Add the metals examined.

.

Point 5: line 48: AA must specify the material of the scalpel used.

Response 5. Add type of material.

Point 6: line 51: Give the country of production of the balance

Response 6. We add data requested of the line 51, 55, 56, 57, 59, 60, 64.

Point 7: line 74: Did the data obtained respected the assumption of normality

of distribution and homoschedasticity?

Response 7. Concentration records for all four metals were adjusted to the normal distribution type (p < 0.05).

Point 8: line 82: The measurement unit should be reported even for the concentration range.

Response 8. The concentration ranges were calculated only for sampling site.

Point 9: line 51: A percentage of the fish samples above the limits should be given.

Response 9. We add the percentages indicated.

Point 10: lines 93-94: Regarding the correlation I suggest to use the Fulton condition factor.

Response 10. We add Fulton's condition factor in the correlation

Point 11: line 114: A comparison with other benthic fishes could be useful to have an epidemiological comparison. I suggest.

Response 11. We add suggested comparison

Reviewer 4 Report

The main objective of this work is to study the concentration of Cd, Pb, V and Zn in muscle tissue of Pterois volitans captured in the Veracruzano Reef System National Park. The analyses were carried out by atomic absorption spectrophotometry.

The work is interesting but, in its present form, the manuscript cannot be accepted. Major revisions are required in accordance with the following recommendations.

GENERAL COMMENTS

Authors should justify the choice of metals such as Zn or V. Zn is an essential microelement, necessary for the organism’s life. The choice of V is very strange: are there in the studied area farm or industrial processes that release V in the environment? More toxic pollutants, such as Hg or As could be considered. The introduction does not provide sufficient background about the toxicity of heavy metals considered and the choice of metals. Why authors choose the invasive species Pterois volitans to assess the levels of pollution of PNSAV? What about “native” species? The presentation of results is bad-wrote, and the discussion is scarce. Authors should make a deep revision of this part. Pay attention to the significant figures. The general rule is that experimental uncertainty (Standard Deviation) should be rounded to one significant digit. In Table 1, such as along the text, authors reported too many digits. For example, replace “0.0326±0.0034” with “0.033±0.003”, and so on. Please correct. Finally, there are many grammatical and spelling errors through the whole manuscript. Considerable improvement should be carried out and an English language check is necessary.

SPECIFIC COMMENTS

Abstract

Lines 10-12, fishes were captured in April 2018, as underlined in line 38. Authors wrote “were analyzed in April 2018”. Please correct.

Lines 13-14. Authors must report only the significant figures.

Materials and methods

Line 43, replace “4° C” with “4 °C”

Line 51, company, city and country of the ADAM analytical balance.

Line 52: sterilized?

Line 54: how authors reach “dry weight” of the samples? 24h to 60 °C is not enough to reach the dry weight. In general, a freeze-drying process is used, that allows a complete loss of water at low

temperature and at low pressure.

Line 56: the description of the digestion method is not accurate. Ramp? holding time?

The description of sample treatment and sample digestion should be improved. See for example Determination of Hg in Farmed and Wild Atlantic Bluefin Tuna (Thunnus thynnus L.) Muscle

Anna Annibaldi, Cristina Truzzi, Oliana Carnevali, Paolo Pignalosa, Martina Api, Giuseppe Scarponi and Silvia Illuminati, Molecules, 2019, 24, 1273; doi:10.3390/molecules24071273.

Line 60: company, city and country of the vacuum pump.

Line 66: the sentence “using the atomic absorption method” is not necessary

Line 68,69, replace “Stand Ards” with “Standards”

Line 70: the concentration of each metal was…

Line 73: replace “was made” with “was studied”

Line 74-76: which statistical software authors used?

Results and discussion

Line 78: was from 16 to 36 cm (mean 23…)

Line 78: was from 32 to 717 g (mean 236±210)

Line 78: the total average concentration and minimum and maximum values of metals was as follow:

Line 80-82: to many figures.

Line 87: what the sentence “limits are indicated by the standard as contaminants” means?

Line 88: “but only Pb records are at risk”, please reformulate the sentence

Lines 87-91: reformulate the sentences

Line 93:” the regression analysis” instead of “the correlation coefficient analysis”

Line 96: the bioaccumulation factor is measured as the ratio between metal concentration in fish and metal concentration in water. Authors can not talk about bioaccumulation considering only the non-relation between heavy metal concentration in muscle and the weight or size of fish.

Line116-118: is the sentence not completed?

Table 1

I suggest inserting law limits for Cd and Pb

Author Response

Response to Reviewer 4 Comments

Thanks for the comments and observations, and at the suggestion of the reviewers, different sections were rewritten and the English language was revised.

Point 1: Lines 10-12, fishes were captured in April 2018, as underlined in line 38. Authors wrote “were analyzed in April 2018”. Please correct.

Response 1. It was corrected as suggested

Point 2: Lines 13-14. Authors must report only the significant figures.

Response 2: It was corrected as suggested

Point 3: Line 43, replace “4° C” with “4 °C”.

Response 3. As suggested.

Point 4: Line 51, company, city and country of the ADAM analytical balance.

Response 4. We add data requested of the line 51, 55, 56, 57, 59, 60, 64.

.

Point 5: Line 52: sterilized?

Response 5. Deleted.

Point 6: line 51: Give the country of production of the balance

Response 6. We add data requested of the line 51, 55, 56, 57, 59, 60, 64.

Point 7: line 74: how authors reach “dry weight” of the samples? 24h to 60 °C is not enough to reach the dry weight. In general, a freeze-drying process is used, that allows a complete loss of water at low temperature and at low pressure.

Response 7. Unfortunately, with lyophilizer it was not possible, and we used the oven for a week and not 24 hours. “For the drying part, 10 g of sample were taken in porcelain capsules that were placed in an oven, at 60 °C for a week, until reaching constant weight”

Point 8: Line 56: the description of the digestion method is not accurate. Ramp? holding time?.

Response 8. Was corrected, “This process was conducted at 150 °C for 5 minutes and 190 °C for 10 minutes, following the manufacturer recommendations (CEM Corporation).

Point 9: The description of sample treatment and sample digestion should be improved. See for example Determination of Hg in Farmed and Wild Atlantic Bluefin Tuna (Thunnus thynnus L.) Muscle.

Response 9. We appreciate the suggestion.

Point 10: Line 60: company, city and country of the vacuum pump.

Response 10. indicated above, No 6.

Point 11: Line 66: the sentence “using the atomic absorption method” is not necessary.

Response 11. We appreciate the observation

Point 12: Line 68, 69, replace “Stand Ards” with “Standards”.

Response 12. As suggested.

Point 13: Line 70: the concentration of each metal was….

Response 13. Was corrected.

Point 14: Line 73: replace “was made” with “was studied”.

Response 15. As suggested.

Point 16: Line 74-76: which statistical software authors used?.

Response 16. Statistical analyses were performed using the PAST software, Version 3.14 [23].

Point 17: Line 78: was from 16 to 36 cm (mean 23…), Line 78: was from 32 to 717 g (mean 236±210)

Response 17. As suggested.

Point 18: Line 78: the total average concentration and minimum and maximum values of metals was as follow:.

Response 18. As suggested.

Point 19: Line 80-82: to many figures.

Response 19. Was corrected.

Point 20: Line 87: what the sentence “limits are indicated by the standard as contaminants” means?

Response 20. Was corrected.

Point 21: Line 88: “but only Pb records are at risk”, please reformulate the sentence.

Response 21. The sentence was rewritten.

Point 22: Lines 87-91: reformulate the sentences.

Response 22. The sentence was rewritten.

Point 23: Line 93:” the regression analysis” instead of “the correlation coefficient analysis”

Response 23. As suggested.

Point 24: Line 96: the bioaccumulation factor is measured as the ratio between metal concentration in fish and metal concentration in water. Authors can not talk about bioaccumulation considering only the non-relation between heavy metal concentration in muscle and the weight or size of fish.

Response 24. Deleted.

Point 25: Line116-118: is the sentence not completed?

Response 25. The sentence was rewritten.

Point 26: Table 1. I suggest inserting law limits for Cd and Pb

Response 26. Was added in the table.

Round 2

Reviewer 2 Report

The comments are  in text

Author Response

Dear Reviewer 2,

We appreciate the observations and comments to our work

below, we add response to them.

Line 19, dry or wet weight?,

dry weight.

Line 36, “y”,

replaced by “and”.

Line 90, I do not understand “y”

replaced by “and”.

Line 106, insert units,

mg kg¹

Line 113, Is this necessary?

It was part of our data analysis

Line 136, The results in the table are expressed in units of dry matter, while the permissible levels in units of fresh weight. Will the lead content exceed the allowable limits after converting the results to fresh mass ?

If, in effect, with the conversion of dry weight to wet weight the concentrations change, and were added to ms (… In this study, the average concentration of cadmium (0.007 mg kg¹wet weight) and lead (0.15 mg kg¹wet weight) ….)

Line 140, According to FAO / WHO in CODEX STAN 193-1995, the allowable limit for lead is 0.3 mg / kg fresh weight, not 0.5 mg / kg (see Table, page 34).

National and international limits were added

Line 143, SD, explanations under the Table.

Was added

In the Table (page 32) in the document FAO / WHO in CODEX STAN 193-1995 I did not notice the permissible level of cadmium for fish. Please check this. Is it 0.05 mg / kg fresh weight?

Was added

Insert units

Was added

Reviewer 3 Report

The Authors replied with satisfaction at all my comments and suggestion.

The MS can be accepted as it is

Author Response

Dear Reviewer,

We appreciate the observations and comments to our work

Reviewer 4 Report

The main objective of this work is to study the concentration of Cd, Pb, V and Zn in muscle tissue of Pterois volitans captured in the Veracruzano Reef System National Park. The analyses were carried out by atomic absorption spectrophotometry.The authors answered almost all the requests for revision, but there are still some points to be clarified. Then, in its present form, the manuscript cannot be accepted. Minor revisions are required in accordance with the following recommendations.

GENERAL COMMENTS

Pay attention to the significant figures. The general rule is that experimental uncertainty (Standard Deviation) should be rounded to one significant digit. In Table 1, such as along the text, authors still reported too many digits. For example, replace “0.0326±0.0034” with “0.033±0.003”, and so on. Please correct. The number of digits in the means values must be the same of the standard deviation. Finally, there are still many grammatical errors through the whole manuscript. Please check English language.

SPECIFIC COMMENTS

Line 13: replace “in the samples” with “in the muscle tissue”

Line 15: Authors must report only the significant figures. 7.3±0.6, 0.7±1.2, 0.66±0.07

Line 16: did not exceed

Line 26: Papaloapan discharges from

Lines 69-70: unit about detection limits?

How many measurements did you perform for each sample?

Line 81-85: to many figures. 23±6; 236±210; Pb= 0.66±0.07 mg kg–¹ (0.47-0.74); V= 7.3 ± 0.6 mg kg–¹ (5.9-8.3); and Zn= 0.7 ± 1.2 mg kg–¹ (0.3-7.0)

Line 85: P < 0.05

Line 106: “ANOVA showed no significant differences among the HM concentrations and the collection sites”. This sentence disagrees with some data reported in Table 1. For example, Zn content in Isla Verde is very high with respect to other sites. Moreover, the standard deviation associated to this value is too high with respect to other SD values. The non-homogeneity of the SD excludes the possibility to apply ANOVA test.

Line 120: However, this study recorded..

Table 1

Still too many figures.

North

Pájaros

11

0.033 ± 0.003

0.72 ± 0.02

8.0 ± 0.2

0.37 ± 0.09

Isla de Sacrificios

2

0.012* ± 0.001

0.49 ± 0.03

6.1 ± 0.2

0.9 ± 0.4

Isla Verde

5

0.016 ± 0.004

0.59 ± 0.02

6.7 ± 0.3

2.1 ± 2.7

South

Enmedio

6

0.024 ± 0.003

0.63 ± 0.03

7.00 ± 0.08

0.37 ± 0.02

Anegada de Afuera

6

0.027 ± 0.002

0.69 ± 0.01

7.42 ± 0.09

0.36 ± 0.02

*0.124 become 0.12, not 0.13

Author Response

Dear Reviewer 4,

We appreciate the observations and comments to our work

below, we add response to them.

Line 13: replace “in the samples” with “in the muscle tissue”

Was added

Line 15: Authors must report only the significant figures. 7.3±0.6, 0.7±1.2, 0.66±0.07

Changes were made as suggested by the reviewer

Line 16: did not exceed

Was added

Line 26: Papaloapan discharges from

Was added

Lines 69-70: unit about detection limits?

Was added

How many measurements did you perform for each sample?

We perform 30 measurements in duplicate for each heavy metal

Line 81-85: to many figures. 23±6; 236±210; Pb= 0.66±0.07 mg kg–¹ (0.47-0.74); V= 7.3 ± 0.6

mg kg–¹ (5.9-8.3); and Zn= 0.7 ± 1.2 mg kg–¹ (0.3-7.0)

Yes, there are many figures, the figures were revised and reduced as suggested.

Line 85: P < 0.05

Was reviewed

Line 106: “ANOVA showed no significant differences among the HM concentrations and the collection sites”. This sentence disagrees with some data reported in Table 1. For example, Zn content in Isla Verde is very high with respect to other sites. Moreover, the standard deviation associated to this value is too high with respect to other SD values. The non-homogeneity of the SD excludes the possibility to apply ANOVA test.

It is right, sorry for the mistake, we checked all figures and all calculations, and found a wrong record and it was already corrected in the results table.

Finally, there are still many grammatical errors through the whole manuscript. Please check English language.

Thanks for the comments and recommendation, the manuscript was reviewed in all sections and checked by a specialist in the English language